# A Sensor Designed to Record Underwater Irradiance with Concern for a Shark’s Spectral Sensitivity

**DOI:** 10.3390/bios11040105

**Published:** 2021-04-03

**Authors:** A. Peter Klimley

**Affiliations:** College of Biological Sciences, University of California, Davis, CA 95616, USA; apklimley@ucdavis.edu

**Keywords:** irradiance, sensor, spectral sensitivity, orientation, shark

## Abstract

To ascertain how scalloped hammerhead sharks make nightly migrations to their feeding grounds as many as 20 km from their daytime abode, a seamount, a sensor was developed that measured irradiance intensity within the spectral range and sensitivity of the vision of the species. Could the sharks guide their movements by sensing the polarity of irradiation energy radiated from the sun or moon that penetrated into the oceanic depths? Two sensory receptors, cones and rods, are present in the retina of sharks to enable them to see both during daytime and nighttime. The peak sensitivity of the cones is red-shifted due to the presence of these wavelengths during the former period, while their response is linear under the range of the high light levels also present at this time; the peak sensitivity of rods is blue-shifted due to the presence of these wavelengths during dawn, dusk, and nighttime and is linear over the complementary range of low light levels. Spectral response curves for these two receptors were determined for sharks, and an attempt was made to match those of the sensors to the shark’s wavelength perception. The first sensor was matched to the photopic range using a photocell covered with a red-shifted gel filter; the second was matched to the scotopic range using a blue-shifted gel filter.

## 1. Introduction

Vision is of less importance to animal navigation in the marine environment because of its dependence upon irradiance, which is absorbed and scattered rapidly in sea water. While birds in the sky can move directionally by flying toward a visual reference point such as the sun, moon, star pattern, or landmark, fishes can do this only by swimming near the surface through which the sun would appear as a blurry, bright spot or close to the bottom where fishes can move parallel to a ridge or valley. Orientation to the sun may exist at least at shallow depths. Lemon sharks (*Negaprion brevirostris*) altered swimming direction in a shallow lagoon from east to west at sunset and from west to east at sunrise [1]. The shoreward movements of parrot fishes were less directional when their vision was blocked with opaque cups than when not blocked by transparent cups and when the sun was obscured by clouds [2].

It is not known whether fishes might use vision to orient in deeper water, particularly when swimming not far from the surface. The scalloped hammerhead, *Sphyrna lewini*, is an obvious candidate to answer this question. Could the sharks guide their movements by sensing the polarity of irradiation radiated from the sun or moon that penetrates more into the oceanic depths? To find an answer to that question, scalloped hammerhead sharks were tracked by boat while carrying an ultrasonic transmitter outfitted with sensors of swimming depth and irradiance, the latter matched to their photopic and scotopic visual sensitivities. The depths at which a scalloped hammerhead was swimming as well as the irradiance levels it experienced were telemetered back to an on-board receiver that automatically decoded the pulse-interval modulated signals into measurements of swimming depth and irradiance [3,4]. In this manner, it was possible to find out whether the hammerhead might be orienting using underwater irradiance.

Irradiance passes into the shark’s eye through a crystalline lens and focusses on the retina, which possesses two types of photoreceptors, namely rods and cones. “Irradiance” is used here in preference to “light” because it refers to those wavelengths both within and outside the visible spectrum of wavelengths perceptible to humans. The rods are the most common light-sensitive organs in the retina of sharks [5,6]. They are distinguished by their oval base and their extended inner and outer segments. The cones are less common, and they possess a conical base and a thick inner segment conjoined with an outer segment that tapers to a point. A section of the retina is shown for the lemon shark (Figure 1a). The section shows a single cone receptor surrounded by rod receptors on either side. The diagram to the right shows a light-adapted and dark-adapted cone as well as a dark-adapted rod receptor (Figure 1b).

The two receptors have different functions [5,6]. The rods are highly sensitive to quanta of irradiance, and enable the shark to distinguish objects under the low levels of irradiance present during dawn, dusk, and nighttime conditions. They also permit the sharks to see well in the blue-shifted spectrum present in oceanic waters (Figure 2b). The cones are less sensitive to quanta of light and permit the shark to distinguish an object during daytime from its background by its slightly different brightness. They also permit the sharks to see well in the red-shifted coastal waters (Figure 2c). Situated within the outer segments of these receptors are two irradiance-sensitive pigments, rhodopsins and porphyropsins [7]. The rhodopsins, which are present in the rods, are most sensitive to blue-green irradiance varying between 497 and 510 nm that are present in the open ocean (as well as during dawn, dusk, and nighttime in coastal waters) [7]; the porphyropsins, which are present in the cones, are red-shifted [8]. They possess a higher peak absorption at a wavelength of 522 nm, and thus are better suited to absorbing irradiance in the spectral conditions of coastal waters during daytime.

The rods and cones of juvenile lemon sharks have both pigments and are thus adapted for daytime and nighttime irradiance conditions. The sensitivity of these sharks can be determined by threshold experiments, during which the subject is exposed to a wide range of wavelengths with successively decreasing irradiance intensities [10]. Upon experiencing a mild shock, the subject raises its nictitating membrane to cover and protect its eyes. If the two stimuli, irradiance and electricity, are paired a sufficient number of times, the shark will blink its nictitating membrane to the irradiance stimulus alone. The individual can then be exposed to irradiance of varying wavelengths of successively lower intensities to obtain a threshold for each wavelength. The blue-shifted sensitivity under low-irradiance conditions is referred to as scotopic vision; the red-shifted sensitivity in the presence of high-irradiance conditions is called photopic vision (Figure 3). The rod receptors are thought to permit sharks to see under conditions when the pigments in their cones are bleached and are incapable of image detection. The function of the “duplex” retina, which possesses both rod and cone receptors, is to increase the range of irradiance intensities, over which the visual system can function optimally [11]. Since irradiance intensity varies greatly from the bright light present during daytime to the dim light present on a moonless night, it is not surprising that the majority of vertebrates have a duplex retina.

## 2. Methods and Materials

One of the most basic irradiance sensors is the photoresistor, most often referred to as a photocell. Its electrical resistance changes with the number of quanta impinging upon its surface. A light sensitive material, cadmium sulfide, is placed on a non-conductive substrate such as a ceramic. The material is deposited in a zig-zag pattern in order to attain the required power and resistance rating (Figure 4A). 

When electromagnetic energy within a particular wavelength is absorbed by the irradiance-sensitive material, the valence electrons travel across the bandgap into the conduction band (Figure 4B), decreasing the resistance and permitting current to pass through the circuit (Figure 4C). Irradiance intensity varies greatly, over ten powers of ten from dawn to dusk [14]. The majority of photoresistors respond linearly over only five log units. This is also true for the pigments present in the rods and cones in an animal’s retina. The rods possess pigments that are sensitive to low irradiance levels, and the cones respond to high irradiance levels. One way to record this broad range of irradiance intensities is to equip a transmitter with two photoresistors, a more sensitive one with its resistance changing linearly over low irradiance intensities and a less sensitive one with its resistances varying linearly over high irradiance intensities. The different spectral sensitivities, blue shifted during low irradiance levels, and red shifted during high irradiance levels, of an animal’s eye sight can be simulated by covering the photocells with different gel filters. This then simulates the scoptopic night vision and photopic day vision of terrestrial and aquatic vertebrates.

It is a challenge for any single electronic device to record irradiance with a linear response over ten logarithmic units. It was thus necessary to equip the underwater transmitter with two sensors, one that recorded high irradiance levels with a red-shifted sensitivity and a second that recorded low irradiance levels with a blue-shifted sensitivity. The “O” type of cadmium sulfide was used to simulate the spectral sensitivity of the sharks (Figure 5) [15]. Its broad sensitivity range encompassed the spectral ranges of both the shark’s scotopic and photopic sensitivities. Two photocells and gel cells were chosen to accomplish this, and their specifications are given in Figure 6. One photocell (vt-904) was chosen because its resistance changed more at lower irradiance intensities; another (vt-902) was chosen as its resistance changed less at lower intensities [16]. A gel filter (Kodak, 57A) was affixed to the upper surface of the former photocell with maximum transmissions, varying from 67.5% to 67.2% over wavelengths ranging from 510 nm to 530 nm to emulate the scotopic visual sensitivity of the hammerhead shark [17]. A gel filter (59A) was attached, covering the surface of the latter photocell with maximum transmissions, varying from 73.6% to 69.1% over wavelengths ranging from 510 nm to 540 nm to emulate the photopic visual sensitivity of the shark [17].

The sensors were calibrated using a reference light source (PR-2300, Photo Research). It consisted of a current regulated, switching power supply, an elapsed time meter, a tungsten lamp, and an opal glass diffuser. The output level was decremented using neutral density filters, while the output of the sensors, pulse-interval modulated, telemetered pulses, were timed by an automatic decoder.

The spectral sensitivities of six dark- and light-adapted lemon sharks are indicated by the black curves in Figure 7. These are electroretinograms (ERGs) taken from intact sharks. The dark-adapted shark were kept in the dark for 20 min periods; whereas the light-adapted sharks were kept in daylight for 20 min before testing. The solid circles indicate arithmetic means; the vertical lines represent +/− standard errors around the mean. The curves were fitted using a Dartnall nomogram template placed at a maximum wavelength of 501 nm for the dark-adapted individuals and 541 nm for the light-adapted individuals. The spectral energy of the beam used in these experiments was measured with a calibrated radiometer (United Detector Technology). At 520 nm, the maximum quantal flux was 9.38 × 10^12^ quanta/cm^2^/s (which converts to 0.097 uW/cm^2^/s). The points on the two curves were scaled around the 520 nm point, which was normalized at 2.0. This was done to facilitate the comparison of the different spectral ranges regardless of differences in overall sensitivity. The light-adapted thresholds were consistently 3.5 to 4 log units higher than the dark-adapted thresholds. The blue and red curves indicate the matching sensitivity curves of the irradiance sensors. These were modified with resistors to permit inter-spectral comparison, but the overall sensitivities of photopic and scotopic sensors were different as were the thresholds of the light- and dark-adapted sharks.

Note that the sensitivities were normalized to emphasize spectral differences, with the dark-adapted measurements being for intensities’ multiple log units less than the light-adapted measurements. The lemon shark’s maximum sensitivity in a spectral curve is evident with a peak at 530 nm if kept in the dark prior to the threshold experiment. On the contrary, a shark kept in daylight has a broadened sensitivity range that is shifted 21 nm toward the red with a peak sensitivity of 541 nm. This effect is termed the Purkinge shift [18,19].

The two sensors were affixed to the upper side of an endcap on an ultrasonic transmitter attached to a shark that was tracked during its nightly foraging excursion to a distance of 20 km from its daytime abode, a seamount, in the Gulf of California. A picture of a prototype transmitter is shown without the gray paint used to match it to the dorsal coloration of the shark in order to see its contents (Figure 8). Other sharks within the schools have been observed to bite off transmitters painted with white bands, used for identification, as they reflect light when the shark accelerates and may be considered potential prey (pers. commun., author). The depths at which a scalloped hammerhead was swimming as well as the irradiance levels it experienced were telemetered back to an on-board receiver that automatically decoded the pulse-interval modulated signals into measurements of swimming depth and irradiance [4]. In this manner, it was possible to find out whether the hammerhead might be orienting using underwater irradiance.

## 3. Results

In order to provide a better understanding of how the field experiment was conducted, the track of the shark is shown in Figure 9 (taken from Figure 5 in [3]) with the locations where a temperature and irradiance profiler was lowered in the vicinity of the shark. Two vessels were required to conduct this experiment, one to track the shark, obtaining its geographic location, and another to describe the photic environment in which the shark was swimming. This required the slow lowering of the unit (a transmitter with the same irradiance sensors) at the end of cable with an electric fishing reel from one boat to record measurements of both variables from which to construct the iso-contours shown in Figure 10. In order to keep up with the rapidly moving shark, the unit could only be lowered to a depth of 200 m. The shark was tagged at Las Animas Island in the morning and made a circular loop throughout the night, exhibiting swimming movements that were highly oriented. In each circle were ten successive directional headings, measured with a sensor in the transmitter affixed to the shark (for design, see [13]). The shark swam in a highly oriented manner much of the night. Note that the r values, a Rayleigh coefficient (see statistical description in [20] of 1.0 indicating swimming in a straight line, at roughly at 17:00 and 18:00 hrs were 0.997 and 0.990, respectively, and later at 23:00 hrs were also non-random with a r value equal to 0.543.

Were the levels of irradiance at the deep depths at which hammerhead sharks swam high enough for perception? If they were, did the irradiance originate from a celestial body, or alternatively did it originate near the shark from bioluminescent organisms living at the depths at which the sharks swam? During nighttime, the intensities of photopic and scotopic irradiance measured near the shark were very diminished, <0.0001 and <0.001 µW cm^−2^ s^−1^ at depths ranging from 150 to 200 m, the uppermost limit to the diving excursions of the scalloped hammerhead shark (Figure 10, taken from Figure 6 in [3]).

One can estimate the irradiance levels present when the shark swam in an oriented manner even below 200 m from Figure 10. One can roughly estimate this by calculating the difference in depth between two irradiance iso-contours. With regard to photopic irradiance, when the shark descended to a depth of 300 m at 18:45 hrs, the irradiance decreased from 0.001 to 0.0001 uW/cm^2^/s over an increase of 25 m in depth, giving a single log unit decrement per 25 m. When the shark dived a depth of 400 m 23:30 hrs, the irradiance decreased from 0.001 to 0.0001 uW/cm^2^/s over a depth increment of 50 m. In the former case, the irradiance would decrement 4 log units over the 100 m depth increase to where the shark swam; in the latter case, the irradiance would also decrement 4 log units over the 200 m increase in depth. The irradiance levels at the depth the shark was now swimming, given attenuation of surface irradiance with depth, would be much below the visual sensitivity of the shark determined in the laboratory.

With regard to scotopic irradiance, when the shark dived to a depth of 400 m at 23:30 hrs, the irradiance decreased from 0.01 to 0.001 uW/cm^2^/s over an increase of 100 m, giving a single log decrement for each increasing 100 m in depth. The scotopic irradiance would decrement 2 more log units over the 200 m further increase in depth. The irradiance level at 400 m would be 0.00001 uW/cm^2^/s, which is lower than the lowest threshold measured for sharks.

Were the levels of irradiance at the great depths at which hammerhead sharks swam high enough for detection? The levels of photopic and scotopic irradiance during nighttime were <0.0001 and <0.001 µW cm^−2^ s^−1^ at depths of 150 to 200 m. These were the shallowest depths to which the scalloped hammerheads rose during their nightly vertical diving excursions (see Figure 10a,b). The levels recorded by the sensors were under the threshold levels of 2.263 and 0.115 µW cm^−2^ s^−1^ for 0.02 s flashes of monochromatic light of 451 and 533 nm. These were the wavelengths to which lemon sharks were most sensitive after light- and dark-adaptation [21]. Lower thresholds for perception of white irradiance were recorded for initially light-adapted lemon sharks after increasing periods of dark adaptation [10]. The threshold sensitivity recorded decreased from 2.45 µW cm*^−2^* s^−1^ after 2 min to an intensity of 0.00024 µW cm^−2^ s*^−^*^1^ after 60 min, with a temporal variability of only one power of ten. The latter sensitivity exceeded by one power of ten the scotopic irradiance level measured at the upper limit of the diving excursions of the scalloped hammerhead shark (Figure 10b). However, if one takes into account the two-log unit additional decrease in irradiance over the 200 m to the 400 m increase in depth, at which the shark was swimming, the irradiance level would be 0.00001 uW/cm^2^/s, which is considerably lower than the lowest threshold measured for sharks. The white light on which these thresholds were based was composed of electromagnetic energy spread over a wide range of wavelengths, unlike the narrow bandwidths of irradiance matching the photopic and scotopic spectral sensitivity curves used to calibrate the sensors on the transmitters. Although not enough irradiance was present for photopic vision, an adequate amount of irradiance may have been present in the blue region of the spectrum for limited scotopic vision at the upper parts of the diving excursions. However, during the shark’s deepest dives, the irradiance from the surface likely diminished to a level below the shark’s perceptual capability.

## 4. Discussion

As mentioned in the Introduction, vision may be less crucial to animal navigation in the marine environment because of its dependence upon irradiance, which is absorbed and scattered quickly in sea water. While birds can fly in a straight line by moving toward a celestial reference point such as the sun, moon, star pattern, or landmark, fishes can accomplish this only by rising to the surface through which the sun is visible as a blurry, bright spot or descending to the bottom where they can swim along a feature in the seafloor such as a ridge or valley. The scalloped hammerhead sharks rarely swam close the surface, so it was unlikely that they were using a blurry image of a celestial object to orient [3].

Even if the sharks were unable to see an object through the surface of the water, it was improbable that even polarized light, emitted from a celestial object, could provide the directional information necessary for the scalloped hammerhead to swim in a straight line at considerable depths. The polarization of irradiance underwater has been observed to be greatest perpendicular in the direction of the azimuth of the sun [22], thus furnishing a potential bearing for the shark to swim toward. Sensitivity to polarized irradiance has been shown in three bony fishes, the halfbeak [23], goldfish [24], and juvenile trout [25]. Even if the sensory ability existed, the amount of polarized irradiance arriving at the shark would be diminished greatly by reflection from particles suspended in the water column.

## 5. Conclusions

The irradiance intensities were less than 0.0001 and 0.001 µW/cm^2^/s at depths of 200 m, respectively, at times when the sharks were swimming in a highly oriented manner with an r value for 10 directional measurements of 0.990 (see Figure 9). However, when one factors in the four-log unit decrease in photopic irradiance over increased depth of 100 and 200 m based on the irradiance extinction, it is doubtful that any irradiance originating from the surface would be present at its swimming depth of 400 m at 23:30 hrs. Furthermore, if one factors in the additional two log unit decrease in scotopic irradiance at the shark, swimming 200 m deeper than the irradiance sonde was lowered, it is doubtful that any irradiance originating at the surface would be at the shark’s depth at 400 m at 23:30 hrs. This indicates that it was highly unlikely the sharks were using vision to find their feeding grounds. It is hoped that this exercise, although historical in nature, leads modern biologists to match the biosensors on their data loggers or ultrasonic transmitters to the sensory and motor abilities of the species that they study. The many sensors currently used to monitor the behavior of animals are reviewed in an article [13] designed to be understandable by biologists as well as engineers.

This paper was written for its heuristic value, that is, for two reasons. Firstly, the paper tries to illustrate the utility of adapting a sensor to emulate perception of a property of irradiance by the visual system of an animal. The same could be done for other sensory systems. Secondly the paper illustrates the utility of a multidisciplinary approach to studying the biology of a species. Few papers have ever been written incorporating the fields of (1) visual anatomy (Figure 1), (2) experimental visual physiology (Figure 3 and Figure 7), (3) electronic engineering and biosensor design (Figure 4, Figure 5, Figure 6 and Figure 8), (4) biotelemetry (Figure 9 and Figure 10), and (5) animal behavior and marine biology (Figure 9 and Figure 10). All of these approaches were used in this study to identify if irradiance from the surface reached the sharks at the depths that they were swimming.

## Figures and Tables

**Figure 1 biosensors-11-00105-f001:**
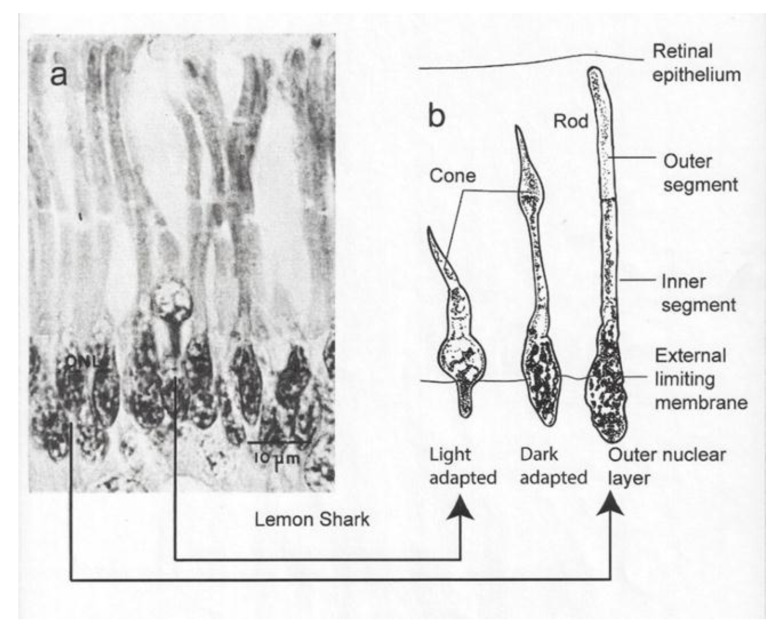
(**a**) Micrograph showing single cone surrounded by rods; (**b**) diagram of dark- and light-adapted cone and dark-adapted rod [5].

**Figure 2 biosensors-11-00105-f002:**
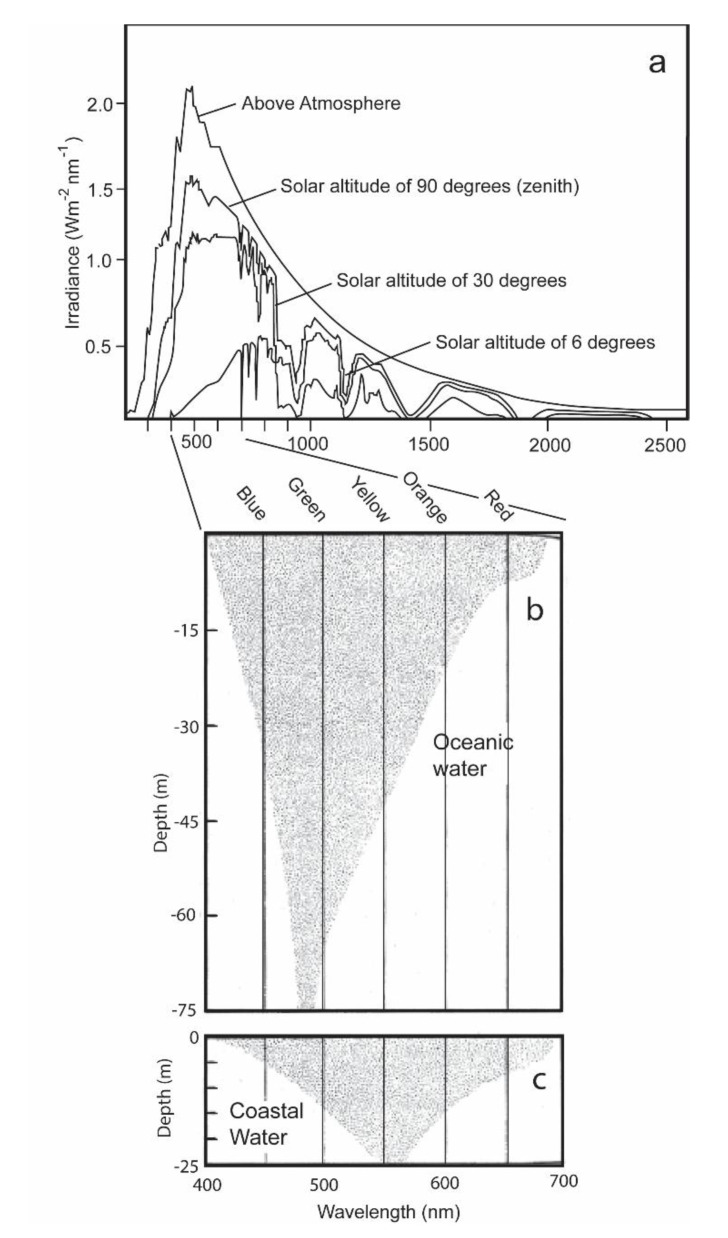
(**a**) Spectral distribution of solar irradiance present from the sun above and within the atmosphere and different solar altitudes with the sun being at its zenith; (**b**) the extinction of daytime irradiance as a function of wavelength with increasing depths in clear oceanic water; (**c**) the extinction of daylight in coastal water with increased turbidity and plankton biomass (**c**) [9].

**Figure 3 biosensors-11-00105-f003:**
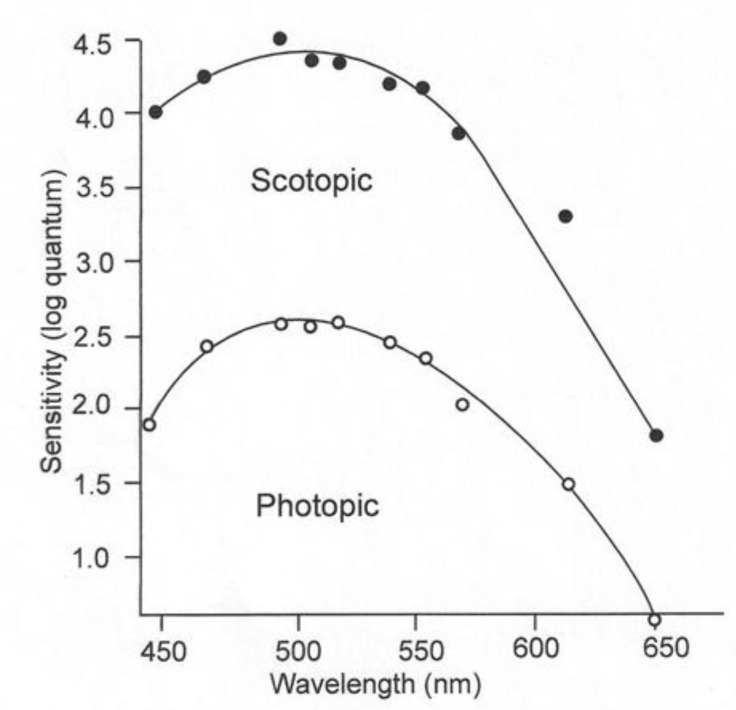
Juvenile lemon shark’s visual sensitivities under low light (scotopic) and high light conditions, based on the response of a single shark [12].

**Figure 4 biosensors-11-00105-f004:**
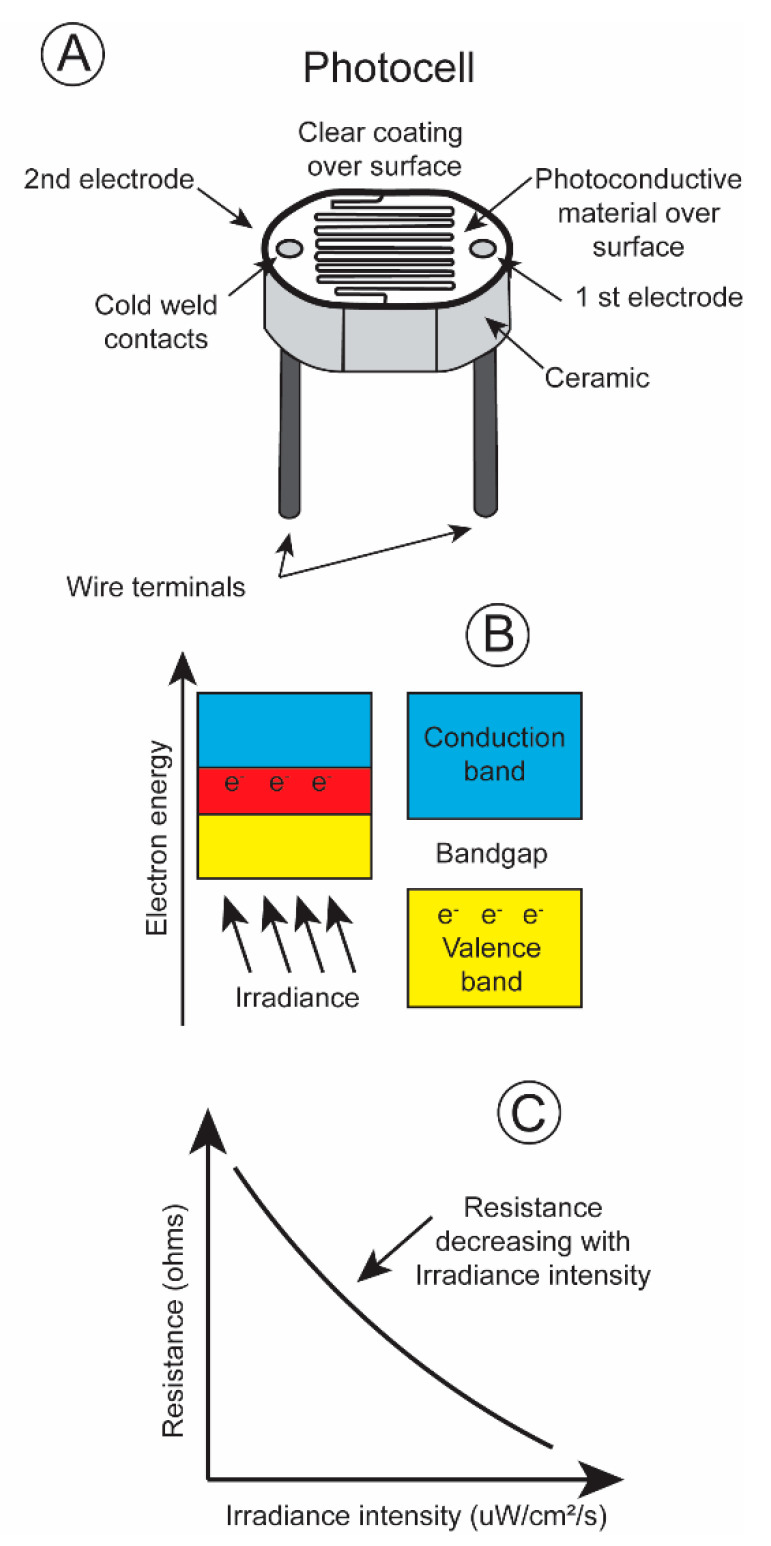
(**A**) Diagram of photocell; (**B**) explanation of operation; (**C**) plot of resistance change as a function of increasing irradiance level [13].

**Figure 5 biosensors-11-00105-f005:**
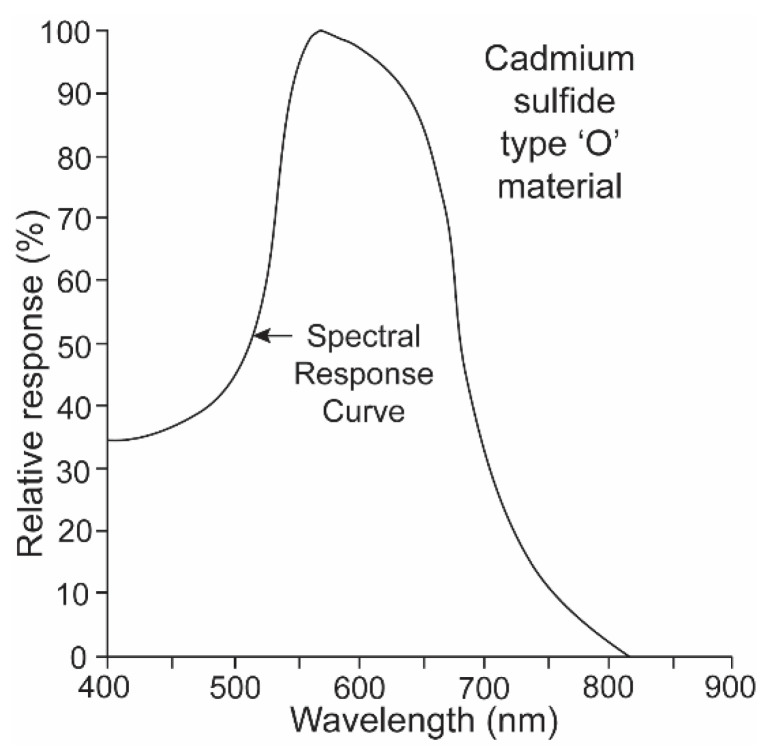
“O” type of cadmium sulfide with a broad spectral range was used to simulate the spectral sensitivity of the sharks.

**Figure 6 biosensors-11-00105-f006:**
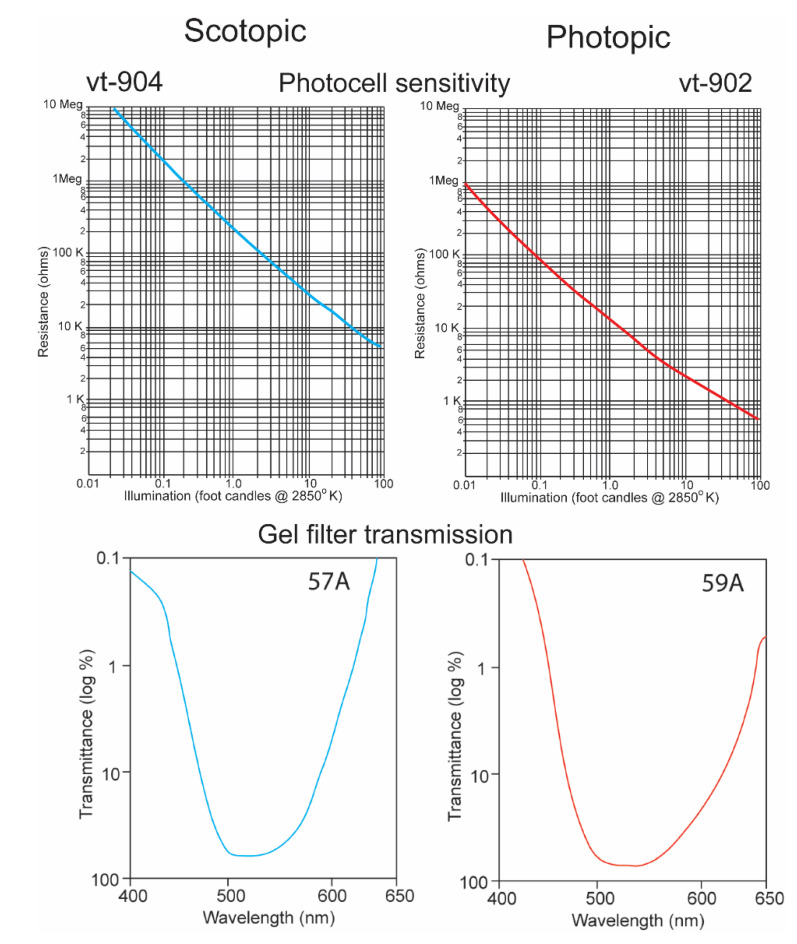
Resistance plotted as function of irradiance for vt-904 and vt-902 photocells (above) [16]. Transmission as function of wavelength for 57A and 59A Kodak gel filters [17].

**Figure 7 biosensors-11-00105-f007:**
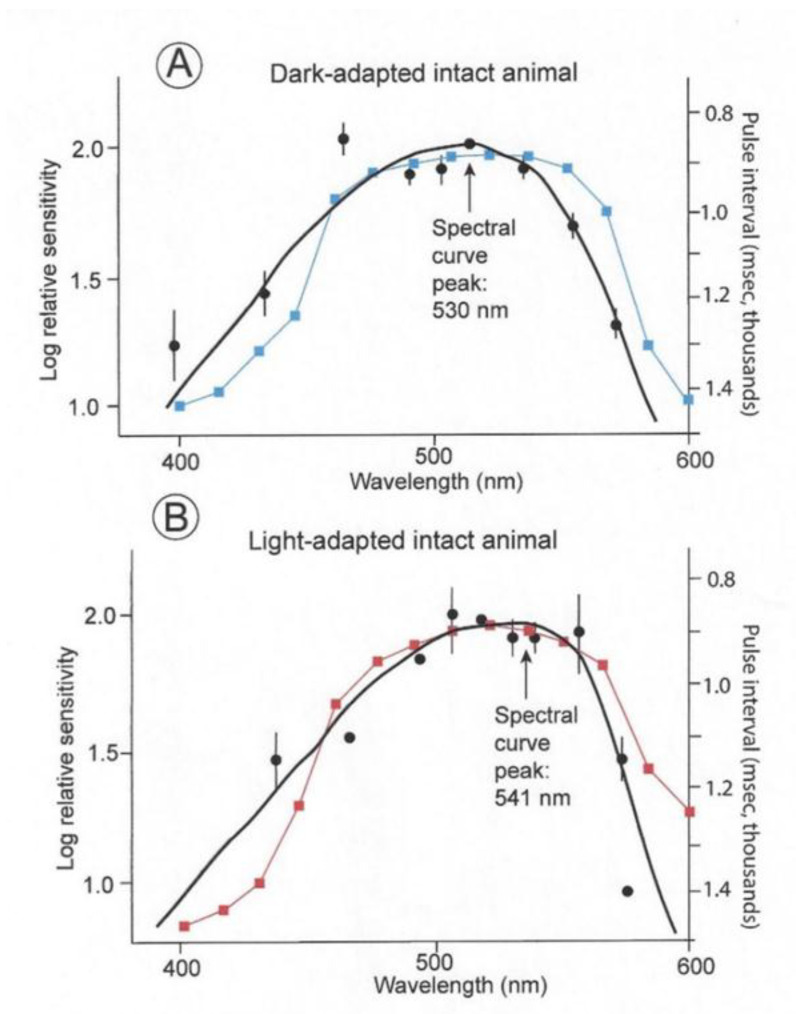
Spectral curves of six dark- and light-adapted lemon sharks (black curves) [18]. Superimposed are the spectral sensitivity curves of the photocells with gel filters used to record measurements of irradiance perceived by dark- (blue) and light-adapted (red) lemon sharks.

**Figure 8 biosensors-11-00105-f008:**
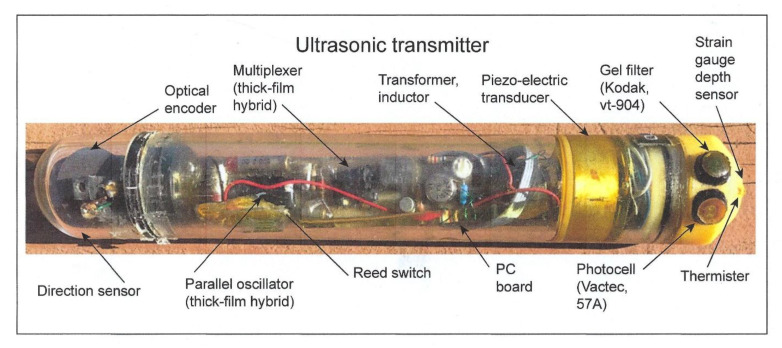
Ultrasonic transmitter with two irradiance sensors. The upper sensor is blue-green from the 47A gel filter to simulate scotopic vision; the bottom sensor is the VT 904 photosensor without the 49A filter used to simulate photopic vision.

**Figure 9 biosensors-11-00105-f009:**
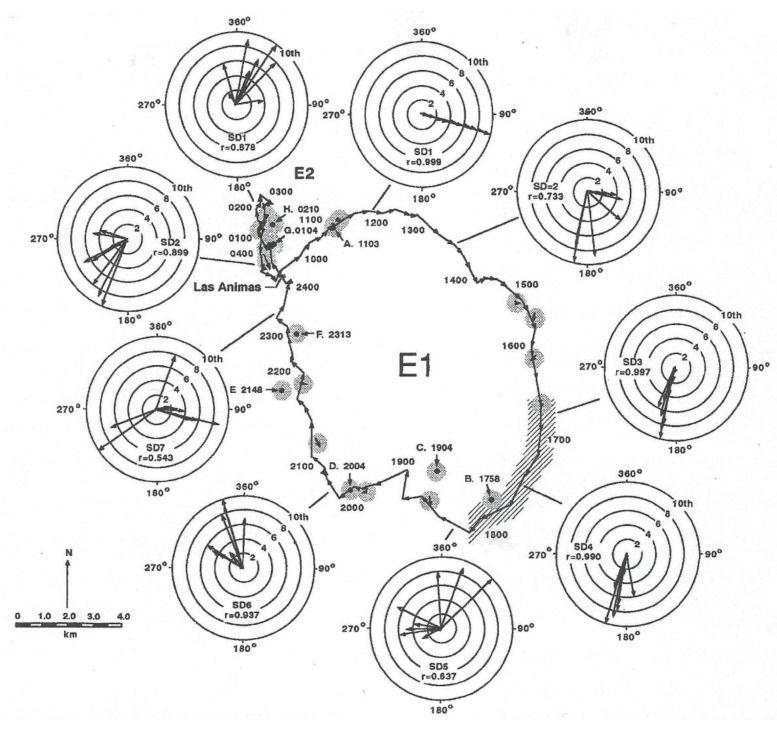
Two homing movements of a 125 m total length (TL) female to Las Animas Island in the Gulf of California on 29 and 30 July 1988. Movement of surface water is indicated by length of arrows in stippled areas between successive positions of a drifter released near the shark. Depth profiles or irradiance and temperature were made at locations identified by letter and time of day (hrs).

**Figure 10 biosensors-11-00105-f010:**
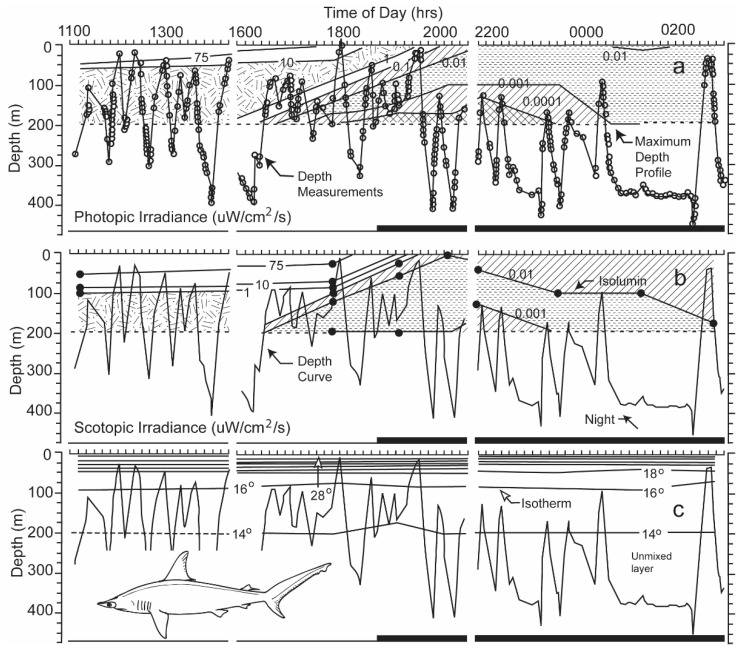
(**a**) The dive profile of a scalloped hammerhead superimposed on contours of irradiance, measured with a sensor emulating the shark’s photopic sensitivity; (**b**) dive profile superimposed irradiance contours emulating its scotopic sensitivity; (**c**) dive profiles superimposed on contours of water temperature [3]. Note that a transmitter, deployed near the tracking boat, could only be lowered to a depth of 200 m to obtain the irradiance and thermal iso-contours.

## Data Availability

Not applicable.

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
