# Peer review of "A Sensor Designed to Record Underwater Irradiance with Concern for a Shark’s Spectral Sensitivity"

_biosensors, 2021, doi:10.3390/bios11040105_

Round 1
Reviewer 1 Report
This paper is more a report of the experiment of the visual capability of the hammerhead shark at the depth the shark migrates during the night.
The outcome of the experiment did not confirm the hypothesis that the shark's visual compatibility could be used for his navigation skills. There is not much scientific novelty besides the convincing proof of the outcome of the experiment, which can be used in further studies of the mechanisms of the shark navigation capabilities. In the paper, the author describes the innovative and complex instrumentation used in the experiments, the procedure, and the method of the experiment. It is evident that the author is an expert working in this field for many years. The paper is interesting for many readers of the journal and it deserves to be published.
To improve the paper may I suggest the following to correct the following.
- Fig 2b and fig 2c add the units for water levels, use the negative sign for levels below the surface
- In all figures where the logarithmic scale is used please use most common marking with actual values i.e. 0.1 1 10 100 etc.
- Fig 4c Please add the scale for both x and y-axis
- Fig 5 (57 A, 59 A) please add units for wavelength.
Figure 8 is from an old paper of yours. It is hard to read. There are missing units on the right y-axis. There are missing explanations so please rewrite this portion of the text.
Reviewer 2 Report
The author presents data showing the performance of a irradiance sensor specifically designed to measure the underwater light field experienced by lemon sharks during the daily cycle. The device was designed taking into account the main features of the view system of this fish group and would be useful to study their behavior. The manuscript is fairly well-written and presents interesting results. However, from the text, it is not easy to distinguish which it was the experimental procedure used by the author to test the device; many Figures and diagrams appear to be direct reproduction of published material and it is confuse from the Discussion how the results obtained with this device can be used in terms of explaining the behavior of the studied species; particularly, it is unclear how the experiments would contribute to assess whether or not the studied species is oriented by using underwater irradiance). Additionally, I find that the references used by the author are out of date, particularly those relative to the biology of shark. Based on these main concerns and the specific comments attached, I think that the manuscript is not useful for publication.
Specific comments
Line 67. It should read 'Figure 2'
Figure 2. Line 82. I wonder if these profiles are representative of the study zone. Optical properties of the water column would be fairly variable. In this sense, I think that it is important to describe the underwater light extinction in the area where the device was tested. Additionally, units for y-axis in panels b and c are missed.
Line 102-103. I wonder if shifts in sensitivity among different individuals would be relevant.
Lines 153-156. The experimental conditions under which this sensitivity experiment was performed should be described, including how the animals were acclimated to dark and light (how long was the acclimation period? what do the author mean with 'light conditions'? How was it supplied). I guess that not only the irradiance but also the light source type would influence on the sensitivity. It will be difficult for other researches to reproduce the experiments (or to compare them with other experimental results) without this information.
Line 158. Figure 6. It is unclear what 'relative sensitivity' means. I suppose that the vertical line on the black points are dispersion measurements (standard deviation?).
Line 165. Please, to define a 'well-lighted environment'.
Lines 168-179. I think that this paragraph should be moved to the section Material and Methods.
Line 184. The environmental conditions under which this experiment was performed should be described, including where, when and how it was carried out. Otherwise, it is difficult that these results are compared with other studies.
Line 201-204. Figure 8. It is confused which data are original and which data have been published previously.
Lines 226-247. This paragraph would be moved almost totally to the Results section.
Line 251. To me, the meaning of 'r' is unclear.
Reviewer 3 Report
In this manuscript, the authors challenge the application of novel sensors to investigate the action of marine organism. These research projects will help to spread the employment of novel devices for bionomics. The manuscript contains original contribution in the field of sensor application as well as bionomics. However, the manuscript still has insufficient explanation of the experimental setup and the research results. The authors should consider the comments on the attached file and revise the manuscript to improve the manuscript before publication in ‘biosensors’.

Round 2
Reviewer 2 Report
I thank the author for considering my comments and modifying the manuscript accordingly. The author's responses have been fairly useful to clear some aspects that (in my opinion) were obscure in the previous version. However, I still find several points that the author should revise before the manuscript being useful for publication:
(1) The author has added some additional information about the experimental procedure (which was also missed by reviewer #3); however, I think that it is still not enough. In my opinion, it is not adequate adding a reference where this information is published. The manuscript has to be self-explicative and the information for the exact experiments shown in Figure 9 and 10 has to be added (at least as Supplementary Material).
(2) I am still confused regarding which Figures and diagrams are a direct reproduction of published material (for instance, Fig1 and 2?), which are modifications from previous material and which are original in this manuscript. Without clear statements about that, it is not easy to see which the originality of this work is.
(3) I will expect that the author update the references (and consequently the Introduction) as well as that the Discussion is contextualized taking into these references. Here you can find two recent articles about view in sharks:
Collin (2018) Clinical and Experimental Optometry https://doi.org/10.1111/cxo.12823)
Tomita et al. (2020; PlosOne, https://doi.org/10.1371/journal.pome.0235342)
Author Response
Reviewer #2
Open Review
(x) I would not like to sign my review report
( ) I would like to sign my review report
English language and style
( ) Extensive editing of English language and style required
( ) Moderate English changes required
( ) English language and style are fine/minor spell check required
(x) I don't feel qualified to judge about the English language and style
|
Yes |
Can be improved |
Must be improved |
Not applicable |
|
|
Does the introduction provide sufficient background and include all relevant references? |
( ) |
(x) |
( ) |
( ) |
|
Is the research design appropriate? |
(x) |
( ) |
( ) |
( ) |
|
Are the methods adequately described? |
( ) |
( ) |
(x) |
( ) |
|
Are the results clearly presented? |
( ) |
(x) |
( ) |
( ) |
|
Are the conclusions supported by the results? |
( ) |
(x) |
( ) |
( ) |
Comments and Suggestions for Authors
I thank the author for considering my comments and modifying the manuscript accordingly. The author's responses have been fairly useful to clear some aspects that (in my opinion) were obscure in the previous version. However, I still find several points that the author should revise before the manuscript being useful for publication:
- The author has added some additional information about the experimental procedure (which was also missed by reviewer #3); however, I think that it is still not enough. In my opinion, it is not adequate adding a reference where this information is published. The manuscript has to be self-explicative and the information for the exact experiments shown in Figure 9 and 10 has to be added (at least as Supplementary Material).
This reviewer request has put the author “between a rock and a hard place”, a “catch 22”. The author was asked by one of the two editors of the Special Issue on Environmental Sensors to write a paper. The author had read the following article on the design of electronic sensors: ‘Whitford, M. and A.P. Klimley. 2019. Biotelemetry made easy: behavioral, physiological, and ecological sensors. Animal Biotelemetry, 7 (26): 1-24; https://doi.org/10.1186/s40317-019-0189-z.’ The second editor advised him to edit the manuscript prior to sending it out for review, giving him the following advice:
“As you states, the paper should be focused on the sensor. I think if this focus is carried through the paper, then the issues with similarities with the previous papers will be elevated and the paper will be suitable for peer-reviewing”.
The figure showing the irradiance levels at the depth the shark swam was simply added to place the description of the design of the sensor in the context of a study to determine the mechanism of orientation of sharks. These results of this study were published in the following article:
Klimley, A.P. 1993. Highly directional swimming by scalloped hammerhead sharks, Sphyrna lewini, and subsurface irradiance, temperature, bathymetry, and geomagnetic field. Marine Biology, 117:1-22.
Reviewer #2 and #3 requested more information on where the irradiance measurements were made relative to the track of the shark. Hence, Figure 9 was added, showing where the sonde that measured irradiance and temperature was lowered throughout the water column during the track of the shark. A seemingly adequate description of the experiment was added to the revised manuscript, but a much longer and more comprehensive description of that particular track and other features of the study is given in Marine Biology, a journal that would be read by marine scientists interested in learning about the migratory pathways of oceanic fishes as well as how they accomplish considerable feats of orientation. The readers of Biosensors are interested in sensor design. Reviewer #3 indicated that the amount of added information is sufficient in his review of the revised manuscript. Note Reviewer #1 had no concern about having more background information about the track. In addition to adding that figure, the author also added a figure on the spectral sensitivity of the photocell in order to provide more on the design of the sensor relative to its use. The author was thus constrained by the editor’s insistence that the paper’s main focus should be on sensor design, as that is the stated scope of the journal, and took exception to satisfy the interests of two of the reviewers.
Based upon the comments of those two reviewers, the author feels the paper has been improved, provided some description of its implementation in a study. If the interests of readers are piqued, they can learn more about the study from the original article, which has little on the design of the sensor but much on its use to eliminate vision as the likely mechanism behind the oriented swimming of the hammerhead – the sharks were shown to swim along magnetic maxima and minima leading from the seamount implicating the electro-magneto sense of the shark. Adding more information as a supplement would be redundant to what has already been published in the scientific literature. Now the two editors of the Special Issue may want to delete those figures, but the author would argue that the interest of Reviewers #2 and #3 justifies the inclusion in the paper.
- I am still confused regarding which Figures and diagrams are a direct reproduction of published material (for instance, Fig1 and 2?), which are modifications from previous material and which are original in this manuscript. Without clear statements about that, it is not easy to see which the originality of this work is.
Note that the origin of Figs. 1and 2 given in the captions. The copyright permissions have been given by the Copyright Clearance Center.
Figure 1. (a) Micrograph showing single cone surrounded by rods; (b) Diagram of dark- and light-adapted cone and dark-adapted rod [5].
- Gruber, S.H.; Cohen, J.L. Visual system of the elasmobranchs: state of the art 1960-1975. In Sensory Biology of Sharks and Rays; Hodgson, E.S., Mathewson, R.F., Eds., U.S. Government Printing Office, Washington D.C., U.S.A., 1978, pp. 11-105.
Figure 2. (a) Spectral distribution of solar irradiance present from the sun above and within the atmosphere and different solar altitudes with the sun being at its zenith; (b) the extinction of daytime irradiance as a function of wavelength with increasing depths in clear oceanic water; (c) the extinction of daylight in coastal water with increased turbidity and plankton biomass (c) [7].
- Drew, E.A. Light. In Sublittoral Ecology; Earll, R., Erwin, D.G., Eds., Clarendum Press, Oxford, England, 1983, pp. 10-57.
The originality of this article is the design of the sensor, and the illustration of the need to match sensors to an animal’s sensory or motor capabilities. These two diagrams from the scientific literature help explain the background, which led to the design of the sensor.
(3) I will expect that the author update the references (and consequently the Introduction) as well as that the Discussion is contextualized taking into these references. Here you can find two recent articles about view in sharks:
Collin (2018) Clinical and Experimental Optometry https://doi.org/10.1111/cxo.12823)
The author thanks Reviewer #2 for making him aware of the Collin article, which is a very comprehensive review of the visual system of sharks and rays. He has incorporated it into his manuscript. Note that it is not a review article, and deals mainly with light and dark adaptation, of which there are not any additional references, but does discuss pigments in a section, and for that reason the paragraph on that has been updated.
Paragraph, lines 62 – 74: updated with references [7] and [8] based on Collin (2018).
Tomita et al. (2020; PlosOne, https://doi.org/10.1371/journal.pome.0235342)
This paper describes the presence of deticles on the eyes of whale sharks, and is judged not germane to the manuscript.
Tomita, T., Murakumo, K., Komoto, S., Dove, A., Kino, M., Kei, M, and Toda, M. 2020. Armored eyes of the whale shark. PLoS ONE 15(6) e03532.
Submission Date
22 February 2021
Date of this review
24 Mar 2021 19:29:44

Reviewer 3 Report
The manuscript was carefully revised and can be accepted.
Author Response
Reviewer #3 is satisfied with my improvements to the manuscript.
Round 3
Reviewer 2 Report
I thank the author for replying my criticisms (my unique objective was contributing to improve the manuscript). I just want to indicate that this reviewer has his own criteria to evaluate a manuscript and that it is not unusual that reviewer's opinions do not match with editor's opinions, although it is obvious that the final decision of accepting the manuscript is a task of the editor.
Regardless of this, I think that the manuscript is useful for publication in the context of the special issue mentioned by the author honestly, I hadn't realized it).